# Associated Impairments among Children with Cerebral Palsy in Rural Bangladesh—Findings from the Bangladesh Cerebral Palsy Register

**DOI:** 10.3390/jcm12041597

**Published:** 2023-02-17

**Authors:** Aditya Narayan, Mohammad Muhit, John Whitehall, Iskander Hossain, Nadia Badawi, Gulam Khandaker, Israt Jahan

**Affiliations:** 1School of Medicine, Western Sydney University, Penrith, NSW 2751, Australia; 2CSF Global, Dhaka 1213, Bangladesh; 3Asian Institute of Disability and Development (AIDD), University of South Asia, Dhaka 1213, Bangladesh; 4Cerebral Palsy Alliance Research Institute, Specialty of Child and Adolescent Health, Sydney Medical School, Faculty of Medicine and Health, The University of Sydney, Sydney, NSW 2006, Australia; 5Grace Centre for Newborn Intensive Care, The Children’s Hospital, Westmead, Sydney, NSW 2145, Australia; 6School of Health, Medical and Applied Sciences, Central Queensland University, Rockhampton, QLD 4701, Australia; 7Discipline of Child and Adolescent Health, Sydney Medical School, The University of Sydney, Sydney, NSW 2006, Australia; 8Central Queensland Public Health Unit, Central Queensland Hospital and Health Service, Rockhampton, QLD 4700, Australia

**Keywords:** associated impairment, comorbidities, register, children, cerebral palsy, low- and middle-income country

## Abstract

Background: We aimed to describe the burden, severity, and underlying factors of associated impairments among children with cerebral palsy (CP) in rural Bangladesh. Methods: This study reports findings from the Bangladesh Cerebral Palsy Register—the first population-based surveillance of children with CP in any LMIC, where children with confirmed CP aged < 18 years are registered by a multidisciplinary team following a standard protocol. Associated impairments were documented based on clinical assessment, available medical records, and a detailed clinical history provided by the primary caregivers. Descriptive analysis, as well as unadjusted and adjusted logistic regression, were completed using R. Results: Between January 2015 and February 2022, 3820 children with CP were registered (mean (SD) age at assessment: 7.6 (5.0) y; 39% female). Overall, 81% of children had ≥1 associated impairment; hearing: 18%, speech: 74%, intellectual: 40%, visual: 14%, epilepsy: 33%. The presence of a history of CP acquired post-neonatally and having a gross motor function classification system levels III–V significantly increased the odds of different types of associated impairments in these children. Most of the children had never received any rehabilitation services and were not enrolled in any mainstream or special education system. Conclusions: The burden of associated impairments was high among children with CP, with comparatively low receipt of rehabilitation and educational services in rural Bangladesh. Comprehensive intervention could improve their functional outcome, participation, and quality of life.

## 1. Introduction

Cerebral palsy (CP) refers to a heterogeneous group of disorders, characterized by motor deficits and functional difficulties such as sensory and cognitive impairment, caused by non-progressive but permanent insults to the developing brain [1]. In the past few years, data from CP registers in low- and middle-income countries (LMICs) have improved the understanding of the epidemiology of CP in LMICs [2,3]. The limited available evidence indicates that the burden of CP is high in LMICs, and most children have severe functional impairment with relatively low access to any rehabilitation services or assistive devices [2,3,4].

Data from high-income countries suggest that children with CP often present with associated impairments [5,6]. In a recent study using data from the Norwegian Patient Registry, it was reported that 95% of children with CP in Norway had at least one comorbidity [5]. A similarly high burden was reported in Australia [6]. Data from the Australian CP Register (ACPR) show that 48% of children with CP had an intellectual impairment, 61% had speech impairment, 12% had hearing impairment, 36% had visual impairment, and 28% had epilepsy [6]. The authors also reported a higher prevalence of associated impairments among children with more severe gross motor function limitations [6]. Although it is expected that the condition among children with CP in LMICs is likely to be worse due to a lack of early intervention and rehabilitation services, there is limited data available on the burden of associated impairments among children with CP in low-resource settings [7,8].

In 2015, the Bangladesh CP Register (BCPR: the first population-based surveillance of children with CP in any LMIC) was established in rural Bangladesh [9]. The BCPR enabled the first population-based estimation of the prevalence of CP (i.e., 3.4 per 1000 children) and provided an understanding of the epidemiology of CP among children in Bangladesh [10]. Furthermore, the data enabled the team to tailor need-based services for children with CP in the country [11,12,13,14].

The presence of associated impairments can complicate the management of children with CP, especially in LMICs, which often lack the necessary resources required to adequately manage the additional burden of disease, leading to negative impacts on health and functional outcome. Estimation of the burden of associated impairments and identification of children more vulnerable to associated impairments could support the development of comprehensive programs and guide adequate resource allocation/mobilization and needs-based service provision for children with CP in LMICs. In this study, we aimed to describe the burden, severity, and underlying factors of associated impairments among children with CP in rural Bangladesh.

## 2. Materials and Methods

The BCPR is an ongoing population-based surveillance of children with CP living in rural areas of Bangladesh [9,10].

### 2.1. Study Participants and Study Area

The study participants are children with confirmed CP aged < 18 years, registered in the BCPR as per the strict clinical definition adopted from the Surveillance of Cerebral Palsy in Europe (SCPE) and Australian Cerebral Palsy Register (ACPR), and a standard protocol published previously [9,15,16]. Children included in this study were recruited from 18 sub-districts of Bangladesh. The study area covers approximately 803,320 households, with an approximate total population of 3,492,088 and a child population of 1,416,254 [17].

### 2.2. Screening and Identification of Children with CP from Community

BCPR uses the key informant method (KIM) to identify children with suspected CP from the community [9,10]. KIM is a valid and widely practiced method that involves the capacity development of local community volunteers known as key informants (KIs) to identify children with suspected CP in their communities via structured training [18]. Following training, the KIs are given 4–6 weeks to identify and list children with suspected CP in their communities [18]. Once identified, the KIs with support from the community mobilizers (paid project staff) invite/bring these children with their caregivers/families to the nearest medical assessment camps for a confirmed diagnosis, registration in the BCPR and services [9,10,18]. More details are available in our previous publications [9,10]. Since its establishment, the BCPR team has trained over 2000 KIs in the surveillance area.

### 2.3. Multi-Disciplinary Medical Assessment

Children with suspected CP identified by the KIs underwent a thorough neurodevelopmental assessment, performed by a multidisciplinary medical assessment team comprising of a paediatrician, a physiotherapist, a counsellor, and a nutritionist. Detailed information about socio-demographic characteristics, pre-, peri and postnatal characteristics, clinical characteristics, nutritional status, rehabilitation, and educational status were documented using the BCPR registration form. The detailed medical assessment procedure is available in our previous publications [9,10] and has been briefly discussed in the subsequent sections.

#### 2.3.1. Socio-Demographic Characteristics

The mothers or primary caregivers accompanying children with CP were interviewed, and the responses were documented following standard reporting guidelines of the census and the demographic and health survey in Bangladesh [17,19].

#### 2.3.2. Known Risk Factors

Data on selected known risk factors of CP were documented by reviewing available medical records and obtaining a detailed clinical history provided by the mothers/caregivers. The components of a pre- and perinatal history such as antenatal care practices (ANC), gestational age, birth weight, attendant of childbirth, history of any complications during delivery, history of febrile illness during pregnancy or labour, history of intrapartum-related neonatal respiratory depression (IPR NRD), and early feeding difficulties were documented following the standard guideline. A child was considered preterm if born before 37 weeks of gestation and was considered to have low birthweight if the birthweight was <2500 g [20]. A child was classified as having a history of intrapartum-related neonatal respiratory depression if they failed to cry at the time of birth, experienced delayed onset of breathing (>1 min), or required assistance to initiate breathing (ranging from drying, stimulation, milking the umbilical cord, or mouth-to-mouth breaths) following birth [21]. Probable intrapartum-related neonatal respiratory depression was defined as neonatal respiratory depression among infants born at term without congenital malformations [21].

#### 2.3.3. Predominant Motor Type, Topography, and Motor Impairment Severity

The predominant motor type of CP and topography were assessed at the time of registration following the ACPR and SCPE classification [15,16]. Gross Motor Function Classification System (GMFCS) level and Manual Ability Classification System (MACS) level were used to document motor impairment severity using standard guidelines [22,23].

#### 2.3.4. Associated Impairments and Epilepsy

The presence and severity of associated impairments (i.e., visual, hearing, speech, and intellectual impairments) and epilepsy were primarily documented based on clinical assessment by the paediatrician and review of any previous medical records and a detailed clinical history provided by the caregivers in the absence of medical records as described in the subsequent sections.

Visual impairment: The presence of visual impairment was recorded based on the assessment of visual acuity and functional vision (including counting fingers, perceiving hand motions, and light perception). In addition, the available medical records were reviewed, and a clinical history was obtained from the primary caregivers. The severity of visual impairment was documented as “some impairment” or “functionally blind” [10].

Hearing impairment: Physical examination (both naked eye examination of the external ear and otoscopic examination) was performed to identify any signs of ear discharge, visual identification of any structural defects, impacted cerumen or foreign objects, perforation, or any other abnormalities of the tympanic membrane and for any conditions that may be contributing to hearing loss requiring further evaluation and treatment. Furthermore, distraction testing was performed to identify signs of hearing impairment among young children and a whispered voice test was performed for children who could communicate to identify signs of any hearing loss [24,25]. The child’s response to name calls and perception of other sounds (e.g., loud noises, clapping) was also observed during the assessment. Any previous history of infection and conditions (e.g., pain, drainage) of the ear and use of any hearing loss intervention were assessed thoroughly by asking the primary caregiver and reviewing available medical records. The severity of hearing impairment was further classified as “some impairment” or “bilateral deafness” [10].

Speech impairment: The presence of speech impairment was documented based on speech and language assessment, review of medical records, and clinical history provided by the primary caregivers of children. Receptive and expressive language, conversational speech quality, and naming quality of the children were evaluated. The severity of speech impairment was classified as “some impairment” or “non-verbal” [10].

Intellectual impairment: The presence and severity of intellectual impairment was determined following the definitions/criteria of the Diagnostic and Statistical Manual of Mental Disorders, 5th Edition (DSM-5) [26], which places emphasis on a child’s adaptive functioning and their performance in daily activities/usual life skills. The assessor interviewed the caregivers to determine if the child had any difficulties in adaptive behaviour, conceptualization, daily communication and comprehension, concentration on daily tasks, learning new skills, relationships, and other practical areas of living [26]. If the caregiver responded yes, then the severity of intellectual impairment was documented as follows: (i) mild— “Can live independently with minimum levels of support”, (ii) moderate—“Independent living may be achieved with moderate levels of support, such as those available in group homes”, and (iii) severe/profound—“Requires daily assistance with self-care activities and safety supervision/requires 24-hour care” [26]. Relevant medical records were also reviewed if available [10].

Epilepsy: Diagnosis of epilepsy was made based on the history of tonic–clonic seizures by interviewing the primary caregivers and reviewing available medical records. In consultation with a paediatric neurologist, a child was documented as having active epilepsy if s/he presented with a history of one or more unprovoked seizures in the preceding three months of data collection at medical assessment camps and after the neonatal period [10,13].

### 2.4. Data Management and Analysis

Data management and analyses were completed using R (version 4.2.1). Descriptive (such as mean, standard deviation, proportions with 95% confidence intervals; CI) and bivariate analysis (cross-tabulation with chi-square and Fisher’s exact test as appropriate) were completed to describe the socio-demographic characteristics, the burden of different impairments, and the potential underlying factors of different types of impairments among children with CP in Bangladesh. Unadjusted and adjusted logistic regression (OR and aOR respectively) were completed to control potential confounding factors and identify the predictors of different forms of associated impairments among children with CP in Bangladesh.

### 2.5. Ethical Considerations

Ethics approval for BCPR was obtained from the Human Research Ethics Committee of Bangladesh Medical Research Council (BMRC) (southasia-irb-2014-l-01), Cerebral Palsy Alliance (EC00402; ref no. 2015-03-02), and the Asian Institute of Disability and Development (AIDD) (southasia-irb-2014-l-01). Informed written consent was obtained from the primary caregiver/parents of the children prior to registration in the BCPR. A participant information sheet written in the Bengali language was provided to each of the primary caregivers/parents of children with CP registered in the BCPR.

## 3. Results

Between January 2015 and February 2022, 3820 children with clinically confirmed CP were registered in the BCPR. The mean (SD) age at assessment was 7.6 (5.0) y and 39% female.

### 3.1. Cohort Profile

#### 3.1.1. Socio-Demographic Characteristics

Most children were aged below 10 years (69.4%) at the time of registration into the BCPR. The majority of their parents had received at least some formal schooling (76% of the mothers and 67% of the fathers). Of all, 92% of families had a monthly income less than 15,000 BDT (~146 USD) (mean (SD) monthly family income: 9953 (7301) BDT ~68 (71) USD). Only 29% of the families were living in semi-permanent or permanent houses. Although 98% of the household had access to tube wells for collecting drinking water, nearly half (43%) did not have any access to sanitary toilets. (Table 1).

#### 3.1.2. Predominant Motor Type, Topography and GMFCS Level

Most children in our cohort had spastic CP (80%) and more than two-thirds of them (72%) had spastic bilateral CP. The majority of children also represented severe functional impairment; 71% of children had GMFCS level III–V (Table 1).

### 3.2. Presence of Associated Impairments

Table 2 summarizes the presence of different forms of associated impairments among participating children. Overall, 82% [95% CI 80.5, 82.9] had at least one type of associated impairment. Hearing impairment was documented in 18% [95% CI 17.2, 19.7] children; of them 4% [95% CI 3.5, 4.8] had bilateral deafness. Speech impairment was identified in 74% [95% CI 73.1, 75.8] of children and 41% [95% CI 39.2, 42.3] of them were non-verbal. Overall, 40% [95% CI 38.5, 41.7] of children had confirmed intellectual impairment, and of them, 8% [95% CI 7.5, 9.3] had severe intellectual impairment. Visual impairment was identified in 14% of the children, including 5% [95% CI 4.7, 5.4] of children with functional blindness. Epilepsy was present in 33% [95% CI 31.3, 34.3] of the children and was reported to have resolved by the age of 5 in 9% [95% CI 8.3, 10.2] of children.

### 3.3. Relationship between Predominant Motor Type and Topography of CP, GMFCS Level, and Associated Impairments

Hearing impairment is significantly (*p* < 0.001) higher among children with Ataxia (29%) compared to other predominant motor types (10% among unilateral spastic CP, 21% among bilateral spastic CP, 19% among dyskinetic CP, and 15% among hypotonic CP). The proportion of speech impairment ranges between 73% and 80% among children with bilateral spastic CP, dyskinesia, ataxia, and hypotonia, and is significantly (*p* < 0.001) lower among children with unilateral CP (56%). Visual impairment is present in 17% of children with bilateral spastic CP and 13% of children with dyskinesia. Intellectual impairment is more common among children with bilateral spastic CP (62%), dyskinesia (63%), and ataxia (63%). Furthermore, 39% of children with ataxia and 37% of children with bilateral spastic CP have epilepsy.

All types of associated impairments are more common among children with GMFCS level III–V compared to children with GMFCS level I–II. Among children with GMFCS level I, 7% have hearing impairment, 58% have speech impairment, 5% have visual impairment, 34% have intellectual impairment, and 20% have epilepsy, whereas these percentages are 28%, 91%, 24%, 78%, and 49% (*p* < 0.001) among children with GMFCS level V, respectively. (Appendix A).

### 3.4. Predictors of Associated Impairments

Appendix A and Table 3 show findings from unadjusted and adjusted logistic regression.

#### 3.4.1. Hearing Impairment

The odds of hearing impairment are significantly higher among children who acquired CP post-neonatally (aOR, 95% CI: 2.96 (2.33–3.74)), ataxic children (aOR, 95% CI: 2.48 (1.60–3.82)), and children with GMFCS level III–V (aOR, 95% CI: 2.64 (2.01–3.50)) when adjusted for antenatal care visits, birth attendants, and history of IPR NRD.

#### 3.4.2. Speech Impairment

The odds of speech impairment are significantly higher among children who have a history of IPR NRD (aOR, 95% CI: 2.31 (1.93–2.77)), acquired CP post-neonatally (aOR, 95% CI: 1.43 (1.11–1.86)), and have GMFCS level III–V (aOR, 95% CI: 2.51 (2.07–3.04)). Furthermore, compared to unilateral spastic CP, children with bilateral spastic CP, dyskinesia CP, ataxia, and hypotonic have significantly higher odds of speech impairment when adjusted for other factors.

#### 3.4.3. Visual Impairment

Unskilled birth attendants (aOR, 95% CI: 1.90 (1.43–2.51)), history of IPR NRD (aOR, 95% CI: 1.38 (1.09–1.77)), post-neonatally acquired CP (aOR, 95% CI: 2.27 (1.74–2.94)), and GMFCS level III–V (aOR, 95% CI: 2.58 (1.90–3.55)) are significant predictors of visual impairment among children with CP in the cohort when adjusted for antenatal care visits and predominant motor type or topography.

#### 3.4.4. Intellectual Impairment

Similar to visual impairment, unskilled birth attendants (aOR, 95% CI: 1.80 (1.39–2.35)), history of IPR NRD (aOR, 95% CI: 2.01 (1.64–2.47)), CP acquired post-neonatally (aOR, 95% CI: 3.26 (2.48–4.33)), and GMFCS level III–V (aOR, 95% CI: 2.00 (1.62–2.46)) are significant predictors of intellectual impairment among children with CP in the BCPR cohort. Additionally, when adjusted for other factors, the odds of intellectual impairment are 1.8 times higher among children with bilateral spastic CP, 2.3 times higher among children with dyskinesia, and 2.0 times higher among children with ataxia than children with unilateral CP.

#### 3.4.5. Epilepsy

The odds of epilepsy are significantly higher among children with a history of IPR NRD (aOR, 95% CI: 1.47 (1.23–1.77)), CP acquired post-neonatally (aOR, 95% CI: 1.73 (1.39–2.14)), have bilateral spastic CP (aOR, 95% CI: 1.28 (1.03–1.59)), dyskinetic CP (aOR, 95% CI: 1.63 (1.16–2.26)), and GMFCS level III–V (aOR, 95% CI: 1.95 (1.60–2.38)).

### 3.5. Rehabilitation and Education Status of Children with CP According to the Presence of Different Associated Impairments

More than half (54.3%) of the children with at least one or more associated impairments have never received any rehabilitation. The proportion of children who have ever received any rehabilitation is lowest among children with hearing impairment and is highest among children with epilepsy. Furthermore, among school-aged children, only 6.2% with hearing impairment, 9.7% with speech impairment, 7.5% with visual impairment, 12% with intellectual impairment, and 9.2% with epilepsy are enrolled in any mainstream or special education schools (Table 4).

## 4. Discussion

In this study, we reported population-based data on the frequency and severity of associated impairments among children with CP in rural Bangladesh. We found a substantially high burden of associated impairments among participating children. We also identified several factors that increased the odds of different types of associated impairments among children with CP registered in the BCPR. Unfortunately, the majority of the children with mild to severe associated impairments never received any rehabilitation services and the school enrolment rate was also low among those with multiple associated impairments compared to children with no associated impairments registered in the BCPR.

Speech impairment is the most common form of associated impairment among children with CP registered in the BCPR, followed by intellectual impairment and epilepsy. The findings are consistent with previously reported data from both LMICs and HICs [6,8,27,28,29]. In a population-based study in Uganda, intellectual impairment and epilepsy were reported among 75% and 45% of children in the cohort, respectively [27]. A few other studies in Vietnam, Moldova, and Ethiopia also reported a similarly high burden of speech and intellectual impairments among children with CP, however, those studies were conducted in hospital-based settings [8,28,29]. When compared to other HICs (such as west Sweden, Norway, and Australia), the proportion of severe intellectual impairment is slightly higher in Bangladesh (29%) compared to west Sweden (21–26%) and Australia (22%), but is similar to Norway (31%) [6,30,31]. The proportion of epilepsy observed among children with CP in Bangladesh (33%) is also similar to that reported in western Sweden (33–34%), Norway (28%), and Australia (28%) [6,30,31]. Interestingly, the proportion of severe speech impairment/non-verbal among children with CP is substantially higher in Bangladesh (41%) compared to Australia (24%) or Norway (28%) [6,30,31]. These differences could be attributed to the differences observed in functional impairment severity, motor type, etiology, diagnosis age, and rehabilitation status of children with CP in these countries. Nevertheless, caution should be taken when comparing or interpreting these findings from different countries and regions due to the differences in data collection methods and study populations. More research is required to understand the regional similarities and differences in the profile of associated impairments among children with CP in LMICs such as Bangladesh.

The presence and severity of different types of associated impairments among children in our cohort vary according to their predominant motor type and topography. Speech, intellectual impairments, and epilepsy are more prevalent among children with bilateral spastic CP and dyskinetic CP, whereas hearing impairment is more common among children with ataxic CP. Similar findings were reported in Uganda (an LMIC) [27], Canada, and Sweden (HICs) [32,33], However, in Australia, speech and hearing impairment, and epilepsy were more common among children with dyskinetic CP, as well as among children with bilateral spastic CP [6].

All types of associated impairments are significantly higher among children with GMFCS level III to V. The findings are consistent with population-based data from Australia (i.e., ACPR), Sweden, and Uganda [6,27,33]. One possible explanation for this is that children with more severe motor impairment have suffered a more severe insult to the developing brain, resulting in a higher degree of brain injury that leads to more extensive disruption of white matter pathways, causing an increased number of associated impairments [34].

A positive relationship is observed between the presence of intellectual impairment or epilepsy and a history of IPR NRD in a child registered in the BCPR. History of IPR NRD is a strong indicator for adverse peri/neonatal events such as hypoxic–ischaemic encephalopathy (HIE) in newborns, and the risk is even higher in LMICs such as Bangladesh where homebirths in the absence of medical professionals are commonly observed. A positive relationship between a history of such adverse peri/neonatal events and gross motor function limitation, as well as a high burden of associated impairments, was previously reported in Sweden [33]. However we could not establish this causal relationship in this study and need further exploration.

Interestingly, the odds of different associated impairments, including hearing impairment, are significantly higher among children with postnatally acquired CP. One possible explanation for this may be that the auditory system is less susceptible than the brain to damage from perinatal hypoxia in the absence of associated ischemia. However, such ischemia is likely to be a complication of postnatal insults to the brain, such as cardiovascular instability due to sepsis, or repeated respiratory failures, thus, ensuring a greater association with hearing loss [35].

Our data show that nearly half of our study participants have never received any rehabilitation services. Developing needs-based rehabilitation services (including assistive devices) could improve the functional outcome and participation of these children (i.e., children with CP and other associated impairments) in daily activities. The very low school enrolment rate in this cohort is likely to be attributed to the limited functional capacity and lack of inclusive education system in the country. Similar findings were reported in other LMICs [2]. Such inequality in meeting the basic needs, e.g., health care, education, and inclusiveness, eventually makes these children vulnerable to falling into the vicious cycle of poverty, disability, and inequity in low-resource settings of LMICs.

Despite our considerable effort, the study has several limitations. Firstly, with the absence of adequate medical records and access to advanced diagnostic tools/equipment, it is difficult to apply the adopted case definitions and documentation of severity in a consistent manner for all children, especially for those with severe motor impairment and clinical complications. This may also have led to an underestimation of the true burden and misclassification of severity in very young children or children with severe intellectual impairments. Second, we used the key informant method to identify children with suspected CP from the community. Although KIM is a cost-effective method compared to door-to-door surveys, it is possible that due to a lack of expertise, the KIs may have missed a few milder cases of CP. This may also slightly influence the proportion of gross motor function limitation levels and severity of different associated impairments among registered children. Third, in the absence of medical records, we had to mostly rely on mother/caregiver-reported clinical history to determine the aetiology, timing, and potential causes of the brain injury that led to CP. As a result, we could not establish the confirmed underlying causes of different types of associated impairments among children with CP registered in the BCPR. Finally, as mentioned in the methods, the BCPR is an ongoing surveillance, and we have not achieved complete case ascertainment in some of the surveillance sites. Hence, we could not report the population-based prevalence of different associated impairments among children with CP in rural Bangladesh.

## 5. Conclusions

Associated impairments are commonly observed among children with CP in rural Bangladesh. The presence and severity of different associated impairments are also influenced by the motor and clinical characteristics of CP among children. It is therefore important to undertake a comprehensive approach for early intervention and rehabilitation services to address the needs for both motor function as well as associated impairments among children with CP. Unfortunately, the lack of service provision delays the initiation of early intervention and limits the opportunity of improving the functional capacity of these children and the scope for their participation in daily activities. Capacity development of mid-level rehabilitation workers, innovations to improve accessibility to assistive devices, and undertaking a community-based approach are some potential strategies to improve the situation in the country.

## Figures and Tables

**Table 1 jcm-12-01597-t001:** Cohort profile (socio-demographic characteristics, predominant motor type, topography, and GMFCS level).

Characteristics	n (%), N = 3820
Age in years, n = 3809 ^1^	
0–4	1412 (37.1)
5–9	1231 (32.3)
10–14	836 (21.9)
15–18	339 (8.7)
Gender, n = 3820	
Female	1485 (38.9)
Male	2335 (61.1)
Maternal literacy, n = 3813 ^1^	
No formal schooling	915 (24.0)
Literate	2898 (76.0)
Paternal literacy, n = 3793 ^1^	
No formal schooling	1242 (32.7)
Literate	2551 (67.3)
Monthly family income, BDT (~ USD), n = 3787 ^1,2^	
<15,000 (~<146)	3475 (91.8)
15,000–25,000 (~146–243)	234 (6.2)
>25,000 (~>243)	78 (2.1)
Accommodation type, n = 3764 ^1^	
Temporary shelter (jhupri)	17 (0.5)
Non-permanent (kutcha) house	2654 (70.5)
Semi-permanent (semi-pucca) house	868 (23.1)
Permanent brick (pucca) house	225 (6.0)
Source of drinking water, n = 3809 ^1^	
Other sources	5 (0.1)
Tap water	79 (2.1)
Tube well	3725 (97.8)
Access to sanitation, n = 3758 ^1^	
No toilet facility	68 (1.8)
Non-sanitary latrine	1551 (41.3)
Sanitary latrine	2139 (56.9)
Predominant motor type, n = 3820	
Spastic	3059 (80.1)
Dyskinesia	244 (6.4)
Ataxia	184 (4.8)
Hypotonia	333 (8.7)
Spastic topography, n = 3059	
Unilateral	854 (27.9)
Bilateral	2205 (72.1)
GMFCS level, n = 3782 ^1^	
I–II	1097 (29.0)
III–V	2685 (71.0)

^1^ Missing data; ^2^ 1 USD ~ 103 BDT.

**Table 2 jcm-12-01597-t002:** Presence of associated impairments among children with CP in Bangladesh.

Associated Impairment	n	% [95% CI]
Number of associated impairments, n = 3820		
None	697	18.2 [17.0, 19.5]
1–2	2658	69.6 [68.1, 71.0]
≥3	465	12.2 [11.2, 13.2]
Hearing impairment, n = 3811 ^1^		
None	3109	81.6 [80.3, 82.8]
Some impairment	529	13.9 [12.8, 15.0]
Bilateral deafness	156	4.1 [3.5, 4.8]
Unknown	17	0.4 [0.3, 0.7]
Speech impairment, n = 3818 ^1^		
None	974	25.5 [24.1, 26.9]
Some impairment	1263	33.1 [31.6, 34.6]
Nonverbal	1557	40.8 [39.2, 42.3]
Unknown	24	0.6 [0.4, 0.9]
Visual impairment, n = 3786 ^1^		
None	3219	85.0 [83.8, 86.1]
Some impairment	355	9.4 [8.5, 10.3]
Functionally blind	178	4.7 [4.7, 5.4]
Unknown	34	0.9 [0.6, 1.2]
Intellectual impairment, n = 3629 ^1^		
None	1193	32.9 [31.3, 34.4]
Mild	418	11.5 [10.5, 12.6]
Moderate	735	20.3 [19.0, 21.6]
Severe	302	8.3 [7.5, 9.3]
Unconfirmed/unknown	981	27.0 [25.6, 28.5]
Epilepsy, n = 3812 ^1^		
No	2187	57.4 [55.8, 58.9]
Currently present	1251	32.8 [31.3, 34.3]
Resolved by age 5 years	351	9.0 [8.3, 10.2]
Unknown	23	0.6 [0.4, 0.9]

^1^ Missing data.

**Table 3 jcm-12-01597-t003:** Predictors of associated impairments among children with CP in rural Bangladesh.

Factors	Hearing	Speech	Visual	Intellectual	Epilepsy
Antenatal care visits					
Adequate	Ref	Ref	Ref	Ref	Ref
Inadequate	0.41 (0.34–0.50)	0.94 (0.78–1.13)	0.46 (0.37–0.57)	0.87 (0.72–1.06)	0.95 (0.81–1.12)
Childbirth attended by					
Doctor/midwife	Ref	Ref	Ref	Ref	Ref
Skilled birth attendant/TBA	0.98 (0.79–1.21)	1.34 (1.12–1.61)	0.88 (0.70–1.11)	1.04 (0.86–1.26)	0.67 (0.56–0.79)
Family members	2.09 (1.61–2.71)	1.15 (0.89–1.49)	1.90 (1.43–2.51)	1.80 (1.39–2.35)	1.09 (0.88–1.36)
History of birth-related complications					
No	N/A	Ref	N/A	Ref	Ref
Yes	N/A	1.17 (0.98–1.39)	N/A	1.14 (0.95–1.37)	1.17 (1.00–1.37)
History of IPR NRD					
No	Ref	Ref	Ref	Ref	Ref
Yes	1.13 (0.91–1.41)	2.31 (1.93–2.77)	1.38 (1.09–1.77)	2.01 (1.64–2.47)	1.47 (1.23–1.77)
Timing of brain injury					
Pre and perinatal	Ref	Ref	Ref	Ref	Ref
Postnatal	2.96 (2.33–3.74)	1.43 (1.11–1.86)	2.27 (1.74–2.94)	3.26 (2.48–4.33)	1.73 (1.39–2.14)
Predominant motor type and topography					
Spastic—Unilateral	Ref	Ref	Ref	Ref	Ref
Spastic—Bilateral	1.30 (0.97–1.75)	1.87 (1.51–2.31)	1.30 (0.95–1.79)	1.84 (1.45–2.33)	1.28 (1.03–1.58)
Dyskinesia	1.43 (0.92–2.20)	4.43 (2.87–7.07)	1.16 (0.71–1.88)	2.30 (1.52–3.53)	1.62 (1.16–2.26)
Ataxia	2.48 (1.60–3.82)	1.91 (1.31–2.84)	0.93 (0.52–1.59)	2.01 (1.35–3.01)	1.01 (0.69–1.47)
Hypotonia	1.03 (0.66–1.57)	1.59 (1.16–2.19)	1.30 (0.83–2.03)	1.25 (0.87–1.79)	1.11 (0.80–1.51)
GMFCS level					
I–II	Ref	Ref	Ref	Ref	Ref
III–V	2.64 (2.01–3.50)	2.51 (2.07–3.04)	2.58 (1.90–3.55)	2.00 (1.62–2.46)	1.95 (1.60–2.38)

**Table 4 jcm-12-01597-t004:** Rehabilitation and education status of children with different forms of associated impairments.

Rehabilitation and Education Status	Hearing		Speech		Visual		Intellectual		Epilepsy	
	No	Yes	No	Yes	No	Yes	No	Yes	No	Yes
N	3109	685	974	2820	3219	533	1193	1455	2538	1251
Ever received rehabilitation services	1776 (58%)	273 (40%)	525 (54%)	1522 (54%)	1770 (55%)	251 (48%)	667 (56%)	749 (52%)	1341 (53%)	705 (57%)
*p*-value	<0.001		>0.9		<0.001		0.03		0.044	
Enrolled in mainstream/special schools	534 (19%)	40 (6.2%)	324 (37%)	251 (9.7%)	532 (18%)	38 (7.5%)	339 (31%)	156 (12%)	468 (20%)	106 (9.2%)
*p*-value ^1^	<0.001		<0.001		<0.001		<0.001		<0.001	

^1^ Pearson’s chi-squared test.

## Data Availability

The data presented in this study are available on request from the corresponding author. The data are not publicly available due to ethical considerations.

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
