# Peer review of "Associated Impairments among Children with Cerebral Palsy in Rural Bangladesh—Findings from the Bangladesh Cerebral Palsy Register"

_jcm, 2023, doi:10.3390/jcm12041597_

Round 1
Reviewer 1 Report
This manuscript describes the results of the Bangladesh Cerebral Palsy Register (BCPR). The methodology of the register has been described already in several published papers. The focus of this study is on the associated impairments among children with CP.
Results
P5 S215
Between Januari 2015 and Februari 2022, 3820 children were registered with CP, mean age at assessment 7.6 yrs (SD 5.0), 61% female:
P5 Table 1:
Gender: Female 38.9%, Male 61.1%
The data of the table are much more likely than the text, mistake in the text, should be male?
Table 1
Maternal literacy 76% literate, Paternal literacy: 67.3 literate.
Table 2
Nonverbal: 40.8% (CI 39.2-42.3): West Europe: 25%
Moderate intellectual impairment 20.3%, severe 8.3% West Europe 50%
Epilepsy current present 32.8% West Europe 25%
P7 S 244
Hearing impairment 29% in Ataxic CP, although ataxia in about 2% of the children, 19% among dyskinetic CP. Hearing impairment is associated with jaundice with dyskinetic CP as motor type. Did the ataxic children suffer from encephalitis in the first year of life?
The distribution of motor type and GMFCS is not described for the whole population in this manuscript, I think it is important to add these data for better understanding of the results.
(Epidemiology of cerebral palsy in Bangladesh: a population-based surveillance study
GULAM KHANDAKER1,2,3,4 | MOHAMMAD MUHIT2,3 | TASNEEM KARIM1,2,3 |
HAYLEY SMITHERS-SHEEDY1,5 | IONA NOVAK5 | CHERYL JONES1,6,7 | NADIA BADAWI1,5
This paper mentioned 79.6% spastic CP, 1% ataxic CP, 8.3% dyskinetic CP; 13.5% GMFCS I, 18.3% GMFCS II, 21.6% GMFCS III, 23.7% GMFCS IV and 22.9% GMFCS V: the Bangladesh sample CP children are much more involved than in the West European registry. The changing panorama of cerebral palsy in Sweden. IX. Prevalence and origin in the birth-year period 1995-1998. Himmelmann K, Hagberg G, Beckung E, Hagberg B, Uvebrant P.
Acta Paediatr. 2005 Mar;94(3):287-94. doi: 10.1111/j.1651-2227.2005.tb03071.x)
In the discussion (P10), the differences are mentioned, but a hypothesis about the origin of the differences is not discussed. Is it possible that lesser involved CP children (GMFCS I and II) are easier not recognized by the KI?
For interpretation of the associate impairments, the properties of the sample does influence the results. To my knowledge, the Swedish registry covers the most years, and as there is a national health care, it is likely the most complete register of the population CP children in west Europe.
Half of the CP children never received rehabilitation: the influence on the impairments could be discussed: it is not very likely that the number of blindness and deafness could be influenced. Also, the high percentage of non-verbal children is striking, as most families have more children, so I think more comments about the possible causes of these differences could be valuable.

Author Response
We would like to thank the reviewer for the constructive suggestions and helpful comments on our manuscript titled “Associated impairments among children with cerebral palsy in rural Bangladesh − findings from the Bangladesh Cerebral Palsy Register”.
Please see below our point-by-point response to the respected reviewer’s comments.
Point 1: This manuscript describes the results of the Bangladesh Cerebral Palsy Register (BCPR). The methodology of the register has been described already in several published papers. The focus of this study is on the associated impairments among children with CP.
Results
P5 S215
Between January 2015 and February 2022, 3820 children were registered with CP, mean age at assessment 7.6 yrs (SD 5.0), 61% female:
P5 Table 1:
Gender: Female 38.9%, Male 61.1%
The data of the table are much more likely than the text, mistake in the text, should be male?
Response 1: Thank you for the correction. We apologize for the typo. As the respected reviewer stated, the correct statement would be 39% female. We have now corrected the number in abstract and main results. See line 33 and 216.
Point 2: Table 1
Maternal literacy 76% literate, Paternal literacy: 67.3 literate.
Table 2
Nonverbal: 40.8% (CI 39.2-42.3): West Europe: 25%
Moderate intellectual impairment 20.3%, severe 8.3% West Europe 50%
Epilepsy current present 32.8% West Europe 25%
Response 2: Thank you for sharing the comparable findings. In the study cohort 76% of the mothers and 67% of the fathers received some form of formal schooling. However, when combined (i.e., both mother and father had received some formal schooling), the literacy rate reduced to 60% (n=2277/3820). We agree that the proportion of different associated impairments varied between Bangladesh and West Europe. These regional differences could be due to many factors, such as difference in population characteristics and access to health care and rehabilitation services, differences in case definition adopted for different associated impairments as well as categorization of the severity. We have also observed differences in the burden of different types of associated impairments among children with CP from different countries and regions. We have now elaborated this in the discussion section. See line 339-359.
Point 3: P7 S 244
Hearing impairment 29% in Ataxic CP, although ataxia in about 2% of the children, 19% among dyskinetic CP. Hearing impairment is associated with jaundice with dyskinetic CP as motor type. Did the ataxic children suffer from encephalitis in the first year of life?
Response 3: Thank you for the valuable comment. Unfortunately, due to absence of medical records we mostly had to rely on the detailed clinical history provided by the mothers/ caregivers of the children. Our clinical notes indicate that a large number of children who had ataxic type of CP and hearing impairment had some signs of encephalopathy in the first month of life (n=53/184 ataxic children had hearing impairment, of them n = 30 had a history of some signs of neonatal encephalopathy e.g., difficulty initiating and maintaining respiration, depression of tone or reflexes, abnormal level of consciousness and often by seizures, early feeding difficulty) (Ref 1). However, in absence of supporting medical records we could not establish a confirmed diagnosis, thus cannot report. This is a limitation of our study and we have now added this in the revised manuscript. See line: 417-421.
Ref 1: Khandaker G, Muhit M, Karim T, Smithers‐Sheedy H, Novak I, Jones C, Badawi N. Epidemiology of cerebral palsy in Bangladesh: a population‐based surveillance study. Developmental Medicine & Child Neurology. 2019 May;61(5):601-9.
Point 4: The distribution of motor type and GMFCS is not described for the whole population in this manuscript, I think it is important to add these data for better understanding of the results.
Response 4: Thank you for the valuable suggestion. We have now added the predominant motor type, topography and GMFCS level of study participants in Table 1. See Table 1 and line 227-232.
Point 5: Epidemiology of cerebral palsy in Bangladesh: a population-based surveillance study
GULAM KHANDAKER1,2,3,4 | MOHAMMAD MUHIT2,3 | TASNEEM KARIM1,2,3 | HAYLEY SMITHERS-SHEEDY1,5 | IONA NOVAK5 | CHERYL JONES1,6,7 | NADIA BADAWI1,5
This paper mentioned 79.6% spastic CP, 1% ataxic CP, 8.3% dyskinetic CP; 13.5% GMFCS I, 18.3% GMFCS II, 21.6% GMFCS III, 23.7% GMFCS IV and 22.9% GMFCS V: the Bangladesh sample CP children are much more involved than in the West European registry. The changing panorama of cerebral palsy in Sweden. IX. Prevalence and origin in the birth-year period 1995-1998. Himmelmann K, Hagberg G, Beckung E, Hagberg B, Uvebrant P.
Acta Paediatr. 2005 Mar;94(3):287-94. doi: 10.1111/j.1651-2227.2005.tb03071.x).
In the discussion (P10), the differences are mentioned, but a hypothesis about the origin of the differences is not discussed. Is it possible that lesser involved CP children (GMFCS I and II) are easier not recognized by the KI?
Response 5: Thank you for the valuable comment. We agree with the respected reviewer that although the Key Informant Method is a very cost-effective strategy to identify children with suspected CP from the community specially in low-resource settings like ours (77% case ascertainment rate at 25% cost of door-to-door surveys), one limitation of KIM is that the KIs are local community volunteers from different backgrounds, and due to lack of sufficient expertise, they may have missed some of the younger and milder cases of CP. However, we also need to consider that in absence of adequate access to early diagnosis, intervention, and rehabilitation services as observed in several studies in LMICs (ref 2 and 3), it is likely that children with CP in the rural communities would present severe gross motor functional limitations at the time of registration. Nevertheless, we have now added the limitation of KIM in terms of identification of milder cases of CP in the community in the revised manuscript. See line 412-417.
Ref 2: Al Imam MH, Jahan I, Das MC, Muhit M, Smithers-Sheedy H, McIntyre S, Badawi N, Khandaker G. Rehabilitation status of children with cerebral palsy in Bangladesh: Findings from the Bangladesh Cerebral Palsy Register. PLoS One. 2021 May 3;16(5):e0250640.
Ref 3: Al Imam MH, Jahan I, Muhit M, Hardianto D, Laryea F, Chhetri AB, Smithers-Sheedy H, McIntyre S, Badawi N, Khandaker G. Predictors of rehabilitation service utilisation among children with cerebral palsy (CP) in low-and middle-income countries (LMIC): findings from the global LMIC CP register. Brain Sciences. 2021 Jun 25;11(7):848.
Point 6: For interpretation of the associate impairments, the properties of the sample does influence the results. To my knowledge, the Swedish registry covers the most years, and as there is a national health care, it is likely the most complete register of the population CP children in west Europe. Half of the CP children never received rehabilitation: the influence on the impairments could be discussed: it is not very likely that the number of blindness and deafness could be influenced. Also, the high percentage of non-verbal children is striking, as most families have more children, so I think more comments about the possible causes of these differences could be valuable.
Response 6: Thank you for the valuable comment. We have now edited our discussion section accordingly. See line 339-348.
Thank you.

Reviewer 2 Report
very interesting paper about the burdens of CP in LMIC. Overal very clear paper.
Some small remarks:
Abstract: "the odds of different ..." maybe better "the odds of mutiple ..."?
results: I would like to see the distribution of types of CP in the population and the distribution on the GMFCS to get a better understanding about the population.
Author Response
We would like to thank the reviewer for the constructive suggestions and helpful comments on our manuscript titled “Associated impairments among children with cerebral palsy in rural Bangladesh − findings from the Bangladesh Cerebral Palsy Register”.
Please see below our point-by-point response to the reviewer’s comments.
Response to Reviewer 2 Comments
Point 1: very interesting paper about the burdens of CP in LMIC. Overall, very clear paper.
Response 1: Thank you for the positive feedback.
Point 2: Some small remarks:
Abstract: "the odds of different ..." maybe better "the odds of multiple ..."?
Response 2: Thank you for the suggestion. We agree with the respected reviewer that the odds of multiple impairments increased with severe motor function limitation of children in our cohort. However, we did not measure the association between different variables and the number of associated impairments. Therefore, we could not report this in our abstract. We have now edited the sentence for better clarifications. See line 36.
Point 3: results: I would like to see the distribution of types of CP in the population and the distribution on the GMFCS to get a better understanding about the population.
Response 3: Thank you for the valuable suggestion. As suggested, we have now added the predominant motor type, topography and gross motor function classification system level (GMFCS) of the cohort in the revised manuscript. See Table 1 and line 227-232.
Thank you.
